# Safflower Polysaccharides Alleviate TNBS-Induced Colitis by Modulating Gut Immunity

**DOI:** 10.3390/foods14183199

**Published:** 2025-09-14

**Authors:** Chao Jiang, Furong Zhu, Shabaaiti Aimaier, Liang Zhang, Md Hasan Ali, Furong Fan, Yating Lu, Mengwei Jia, Dongsen Wu, Haipeng Yin, Jianwang Wei, Shenghui Chu, Min Liu

**Affiliations:** Key Laboratory of Xinjiang Phytomedicine Resource and Utilisation, Ministry of Education, Institute for Safflower Industry Research, Pharmacy College, Collaborative Innovation Center for Efficient Safflower Production and Resource Utilization of XPCC, Shihezi University, Shihezi 832002, China; 13918049521@163.com (C.J.); zfr210921@163.com (F.Z.); sabaat_hven@163.com (S.A.); zhangliang@stu.shzu.edu.cn (L.Z.); hasan.yzu@gmail.com (M.H.A.); f721572869@163.com (F.F.); 15979116392@163.com (Y.L.); jmw079@126.com (M.J.); wudongsen@stu.shzu.edu.cn (D.W.); 13565162388@163.com (H.Y.); w16650554611@163.com (J.W.)

**Keywords:** colitis, safflower polysaccharides, intestinal immunity, CHI3L1

## Abstract

This study aims to investigate the potential immunological mechanisms by which Safflower polysaccharides (SPSs) regulate colitis. The therapeutic effect of SPSs on colitis was investigated by trinitrobenzene sulfonic acid (TNBS)-induced rats model, TNF-α-stimulated Caco-2 cells, LPS-induced THP-1 cell model, and a co-culture model of Caco-2 and THP-1. The results demonstrated that SPSs effectively ameliorated clinical symptoms, reduced the expression of pro-inflammatory factors, restored colonic pathological damage, regulated the body’s immunity, and inhibited intestinal macrophage M1 polarization in vivo. In vitro, SPSs could alleviate the inflammatory response of epithelial cells, inhibit macrophage M1 polarization and regulate epithelial-immune cells interaction. Through the experimental study of siRNA-CHI3L1 and r-CHI3L1, it was found that CHI3L1 mediated the interaction between epithelial cells and immune cells. This study demonstrated that SPS can significantly improve clinical symptoms and alleviated colonic damage in TNBS-induced colitis rat models. The underlying mechanisms are associated with STAT3/NF-κB signaling and immunomodulation, where the immunoregulatory effect is based on CHI3L1-mediated epithelial-immune cell interaction mechanisms.

## 1. Introduction

Colitis is an immune-mediated inflammatory bowel disease that begins in the rectal mucosa and extends proximally to involve the colon [1,2]. The primary clinical symptoms in colitis patients include abdominal pain, bloody diarrhea, tenesmus, and increased bowel movement frequency [3]. Without early intervention, colitis may progress to colon cancer, posing a life-threatening risk. Currently available medications for colitis are limited, with conventional treatments including aminosalicylates, corticosteroids, and immunomodulators such as azathioprine and 6-mercaptopurine. These therapies are typically employed as first-line treatments for mild-to-moderate colitis. For patients with moderate-to-severe disease, anti-tumor necrosis factor agents have been widely used, though they often cause significant side effects [4] that impair quality of life. Consequently, there is an urgent clinical need for highly effective natural active compounds with low toxicity and minimal side effects for colitis treatment.

Studies have shown that naturally derived polysaccharides can play an anti-inflammatory role as immunomodulators [5]. Safflower (*Carthamus tinctorius* L.), a dried flower of the Asteraceae plant, is characterized by its pungent, slightly bitter, and warm properties. Safflower not only has a long history of application as a spice, but also its value in the food field has been widely explored. It is widely used in Europe to make edible oil, and its water extract is developed into a natural food colorant because of its bright color. In this study, the safflower polysaccharide (SPS) was used as the research object, focusing on its immunomodulatory effect and providing scientific basis for its application in the field of functional food. SPS is the primary water-soluble macromolecule isolated from safflower filaments, represents one of the core bioactive components responsible for its pharmacological effects. Modern pharmacological research has confirmed that SPS possesses antitumor, antioxidant, and immunomodulatory properties [6]. Specifically, SPS may inhibit the progression of tongue squamous cell carcinoma by regulating the expression of Bcl-2, COX-2, and Bax [7]. Wang et al. reported that SPS-1 significantly suppresses AOM/DSS -induced colorectal cancer by activating macrophage M1 polarization through the NF-κB signaling pathway [8]. A 2022 study highlighted SPS’s potent hydroxyl radical scavenging activity, DPPH radical scavenging capacity, and ABTS radical quenching ability, suggesting its potential as an antioxidant [9]. Previous research from our group demonstrated that SPS markedly ameliorates inflammation in DSS-induced colitis mice, improves intestinal structural integrity, and modulates gut microbiota composition to treat colitis [10]. However, whether SPS alleviates colitis through immunomodulatory mechanisms remains unclear.

Epithelial cells possess the ability to detect abnormalities in the microenvironment, emit reinforcement signals, and secrete defensins [11,12]. Equipped with an array of pattern recognition receptors, these cells can sense and respond to various environmental cues. These signals initiate downstream cascades that shape and direct epithelial stem cell responses, ultimately enhancing barrier integrity, alerting neighboring epithelial cells, recruiting immune cells, and promoting tissue repair. The interplay between intestinal epithelial cells and immune cells plays a central role in maintaining gut homeostasis, preserving barrier function, regulating immune responses, and defending against diseases. This cellular dialog facilitates coordinated responses that maintain physiological balance while strengthening host defense mechanisms. Effective communication between epithelial and immune cells is therefore crucial for preventing infections [13], as their synergistic interaction forms the foundation of intestinal mucosal immunity.

As the largest barrier between the organism and the external environment [14], the intestine serves not only as the primary site for digestion and absorption, but also constitutes the largest mucosal immune system in the body owing to its extensive mucosal surface area and unique structural and functional characteristics. The intestinal immune system achieves local defense and systemic immune regulation through collaborative interactions among the mucosal barrier, gut microbiota and their metabolites, and immune cells [14,15]. Macrophages represent a critically important class of immune cells in animals. Healthy intestines harbor resident monocyte-derived macrophages [16] that have long been regarded as custodians of the intestinal mucosa [17], playing an indispensable role in normal gastrointestinal function and maintaining tissue homeostasis. Characterized by heterogeneity and plasticity [18], macrophages can polarize into M1 (classically activated) and M2 (alternatively activated) phenotypes in biological and pathological contexts [19]. Tissues from colitis patients exhibit a higher proportion of M1 macrophages, and this imbalance in macrophage polarization profoundly influences ulcerative colitis progression [20]. M1 macrophages participate in the pathogenesis of colitis. When pathogens invade the intestine, they first breach the physical barrier and activate subepithelial defense cells, particularly macrophages, which undergo M1 polarization. Activated M1 macrophages secrete large quantities of pro-inflammatory cytokines and chemokines, including IL-6, IL-8, and TNF-α. These cytokines further damage epithelial cells, exacerbating mucosal injury and promoting ulcerative colitis development [21]. Consequently, increasing research focuses on modulating macrophage polarization states to alleviate or treat Colitis.

Chitinase-3-like protein 1 (CHI3L1), a member of the glycoside hydrolase family 18 [22], is a secreted glycoprotein with a molecular weight of about 40 kDa. The human CHI3L1 gene is located on the short arm of chromosome 1 (1p36.13), consisting of nine exons and eight introns, spanning approximately 26 kb in length. CHI3L1 is predominantly expressed in macrophages, neutrophils, cancer cells, endothelial cells, and epithelial cells [23,24], where it plays critical roles in cellular regeneration, migration, and tissue remodeling. As a multifunctional secreted glycoprotein, CHI3L1 broadly participates in inflammatory responses, immune regulation, and cellular signaling pathways, with aberrant expression closely linked to the pathogenesis of various diseases. In inflammatory bowel disease, colonic epithelial cells secrete CHI3L1 into the intestinal lumen, where it mediates pro-inflammatory effects [25]. Our early experimental studies demonstrated that CHI3L1 expression was significantly upregulated in the colonic tissues of DSS-induced colitis. However, the precise molecular mechanisms underlying CHI3L1′s pathological functions in this context remain to be fully elucidated.

In summary, we hypothesize that in colitis, intestinal epithelial cells secrete CHI3L1, which subsequently mediates M1 macrophage polarization, thereby exacerbating colitis pathological progression. To validate this hypothesis, our in vivo experiments demonstrated that SPS exhibits anti-inflammatory effects and inhibits M1 macrophage polarization while suppressing CHI3L1 expression. In vitro studies revealed that SPS can alleviate cellular inflammation and restrain M1 macrophage polarization. Through co-culture systems, we established that epithelial cell inflammation intensifies M1 macrophage polarization, partially mediated by CHI3L1 protein secretion. This study identifies a therapeutic target for colitis patients and provides a theoretical foundation for SPS-targeted CHI3L1 therapy in colitis treatment.

## 2. Materials and Methods

### 2.1. Materials and Reagents

SPS was provided by Shanghai Yeasen Biotechnology Co., Ltd. (Shanghai, China). Trinitrobenzene sulfonic acid (TNBS) and PVDF membrane were purchased from Sigma-Aldrich Co. (St. Louis, MO, USA). Chromatography-grade acetonitrile was obtained from Thermo Fisher Scientific (Waltham, MA, USA). Primary antibodies against STAT3, p-STAT3, CHI3L1, NF-κB p65, and p-NF-κB p65 were supplied by Abcam (Cambridge, UK). Enzyme-linked immunosorbent assay (ELISA) kits for CHI3L1, sIgA, IgG, and IgM were purchased from Elabscience Biotechnology Co. (Wuhan, China). Myeloperoxidase (MPO) Activity Assay kit was purchased from solarbio (Beijing, China).

### 2.2. Chemical Composition Analysis and Average Molecular Weight Distribution Determination

The total sugar, polyphenol, and protein contents were determined using the phenol-sulfuric acid method, Folin–Ciocalteu assay, and BCA method, respectively. The molecular weight of SPS was measured by a gel chromatography system coupled with refractive index detection and multi-angle laser light scattering. The column temperature was maintained at 45 °C with an injection volume of 100 μL. Mobile phase A consisted of 0.02% NaN_3_ and 0.1 M NaNO_3_, delivered at a flow rate of 0.6 mL/min. The retention time was recorded, and the relative molecular weight of the polysaccharide sample was calculated based on the calibration curve equation.

### 2.3. Monosaccharide Composition Analysis of SPS

Using the Shimadzu high-performance liquid chromatography system (Kyoto, Japan), a Symmetry C18 column (4.6 × 250 mm) was employed with a mobile phase flow rate of 0.8 mL/min, column temperature of 30 °C, and injection volume of 10 μL. Mobile phases A and B were PBS buffer (0.1 M, pH 7.4) and acetonitrile (83:17, *v*/*v*), respectively. The detection wavelength was set at 245 nm, with the column temperature maintained at 30 °C and an injection volume of 10 μL. The monosaccharide composition and content of SPS were identified based on the retention times and peak areas of standard monosaccharides.

### 2.4. Establishment of the Rat Colitis Model and Grouping Treatment

Thirty-six eight-week-old male Sprague Dawley rats were purchased from the Experimental Animal Center of Xinjiang Medical University. The rats were housed under controlled conditions at a room temperature of 25 ± 1 °C, humidity of 60 ± 5%, and a 12 h dark/light cycle. After a 7-day acclimatization period, the experiment commenced. The rats were randomly divided into six groups (*n* = 6 per group): control group (Control), TNBS model group (Model), 5-ASA treatment group (5-ASA), low-dose safflower polysaccharide treatment group (L-SPS), medium-dose safflower polysaccharide treatment group (M-SPS), and high-dose safflower polysaccharide treatment group (H-SPS). The model group was given 100 mg/kg 2% TNBS solution by colon administration, and the blank group was given normal saline by colon administration according to the body weight of rats. 5-ASA rats were intragastrically administered with 100 mg/kg/d. The rats in the L-SPS group were given 60 mg/kg/d, the rats in the M-SPS group were given 120 mg/kg/d, and the rats in the H-SPS group were given 180 mg/kg/d (Figure 1). Body weight and the occurrence of loose or bloody stools were monitored and recorded daily. Upon completion of the modeling process, the rats were euthanized, and serum and tissue samples were collected. All experiments were conducted in accordance with the principles outlined in the *Guide for the Care and Use of Laboratory Animals* (National Institutes of Health, Bethesda, MD, USA) and were approved by the Medical Ethics Committee of Shihezi University (A2022-93-02).

### 2.5. Assessment of Disease Activity Index (DAI)

According to Table 1, the DAI was determined by scoring weight loss, loose stool, and bloody stool, with DAI = (weight change score + bloody stools score + loose stools score)/3.

### 2.6. Histopathological Analysis

The distal colon tissue was rinsed with 0.9% saline solution, then fixed in 4% paraformaldehyde solution and embedded in paraffin. The embedded tissue was sectioned into 5 μm slices and stained with hematoxylin and eosin (H&E) for histological damage observation.

For electron microscopy, the colon tissue was immediately placed in EM fixative and fixed for 6 h, then transferred to PBS buffer. Subsequent fixation was performed with 1% acetic acid for 2 h. After dehydration and immersion, the tissue was embedded sequentially in a 40 °C oven for 12 h and a 65 °C oven for 48 h. Ultrathin sections were cut, and the morphology of colon epithelial cells was observed under electron microscopy, with images captured for analysis.

### 2.7. Enzyme-Linked Immunosorbent Assay (ELISA)

The levels of CHI3L1, sIgA, IgG and IgM were detected according to the instructions of the corresponding ELISA kits.

### 2.8. Real-Time Quantitative PCR (qRT-PCR) Analysis

Total RNA was isolated and purified using the standard TRIzol protocol. cDNA synthesis was performed using the PerfectStart^®^ Uni RT&qPCR Kit, with specific primers added for subsequent polymerase chain reaction. The primer sequences utilized are listed in Table 2. GAPDH was used as the internal reference gene. Relative gene expression levels were calculated using the 2^−∆∆Ct^ method.

### 2.9. Western Blot (WB) Analysis

Tissues or cells were added to an EP tube, followed by the addition of lysis buffer (RIPA–PMSF–phosphatase inhibitor = 100:1:1) and incubation at 4 °C for 30 min. The mixture was centrifuged at 4 °C, and the supernatant was collected. Protein concentration was determined using a BCA assay kit (Beyotime, Shanghai, China), after which samples were heat-denatured. Protein separation was performed by SDS-PAGE electrophoresis, followed by transfer onto a PVDF membrane. The membrane was incubated with primary antibody at 4 °C overnight, then subsequently with secondary antibody for 1 h. Imaging was performed using ECL chemiluminescence reagents (Affinity Biosciences, Beijing, China) and a Tanon 5200 Multi automatic chemiluminescence (Shanghai, China) imaging system. Band intensity was quantified by grayscale value analysis using ImageJ 1.46r software (National Institutes of Health, Bethesda, MD, USA).

### 2.10. Flow Cytometry

Macrophages were isolated from rat colonic tissue or THP-1 cells were collected, followed by staining with anti-CD14 and anti-CD86 antibodies. M1-polarized cells were then analyzed by flow cytometry.

### 2.11. Determination of Myeloperoxidase (MPO) Activity

The colon tissue was placed in an EP tube, and an appropriate amount of PBS was added. The mixture was homogenized and centrifuged (10,000× *g*, 10 min, 4 °C), followed by supernatant collection. Subsequent operations were performed according to the manufacturer’s instructions.

### 2.12. Cell Culture and Co-Culture

Caco-2 and THP-1 cells, obtained from Procell Biotechnology Co., Ltd. (Wuhan, China), were maintained in DMEM or 1640 medium, respectively, supplemented with 10% fetal bovine serum and 1% penicillin/streptomycin at 37 °C in a 5% CO_2_ atmosphere. For the co-culture model establishment, THP-1 cells were plated at the bottom of 24-well Transwell chambers (Corning, New York, NY, USA) and M0 macrophage polarization was induced through treatment 24 h with 100 ng/mL PMA. Simultaneously, differentiated Caco-2 monolayer cells were seeded in the upper chambers. Co-culture was conducted with concurrent exposure to 100 ng/mL TNF-α for 24 h prior to cell harvesting for subsequent analyses.

### 2.13. siRNA Transfection

CHI3L1-specific siRNA duplex was transiently transfected into Caco-2 cells using the siRNA-mate plus transfection kit. Following 24 h of intervention, M0-polarized THP-1 cells were co-cultured with the transfected Caco-2 cells, after which THP-1 cells were collected for subsequent experiments. The siRNA sequences used are as follows: Sense strand 5′-GUGCUGCUCUGCAUACAAATT-3′ Antisense strand 3′-UUUGUAUGCAGAGCAGCACTT-5′.

### 2.14. Cell Viability Assay

According to the manufacturer’s cell counting kit 8 (CCK-8) detection protocol, the cells were seeded into 96-well plates at a density of 5 × 10^3^ cells/well, and the CCK-8 reagent was added to incubate for 2 h. Six replicate wells were prepared in each experimental group.

### 2.15. Statistical Analysis

All statistical analyses were performed using GraphPad Prism 8.0 (GraphPad Software, La Jolla, CA, USA), and the data were expressed as mean ± SEM (standard error of the mean).With *t*-tests used to compare between-group data and one-way ANOVA for multiple-group comparisons. *p* values (*p*) < 0.05 were considered statistically significant.

## 3. Results

### 3.1. Physicochemical Properties of SPS

The compositional analysis of SPS revealed that it contained 60.32% total sugar, 2.89% protein, and 1.26% total polyphenols. Molecular weight determination showed an average molecular weight of 31.11 kDa. Monosaccharide composition analysis demonstrated that SPS was primarily composed of glucose (21.63%), galactose (15.60%), rhamnose (2.47%), and arabinose (5.50%) (Appendix A).

### 3.2. SPS Ameliorates TNBS-Induced Colitis in Rats

In the TNBS-induced colitis model, rats in the experimental groups exhibited characteristic disease symptoms including reduced appetite, ruffled fur, progressive weight loss, and occult blood in stool. All TNBS-treated groups (excluding the Control group) showed significant weight reduction starting from Day 1 of the experiment, accompanied by markedly decreased food and water intake. However, rats treated with 5-ASA and various doses of SPS demonstrated a weight recovery trend beginning on Day 3, with a concurrent improvement in nutritional intake parameters (Figure 2A,B and Appendix A).

The Disease Activity Index (DAI) scores of Model group rats exhibited a significant increase, both 5-ASA and SPS treatment groups effectively reduced DAI scores (Figure 2C). Histopathological examination revealed characteristic colon shortening in Model group rats, which was substantially alleviated by medium- and high-dose SPS interventions (Figure 2D and Appendix A). These findings suggested that SPS significantly ameliorates TNBS-induced colitis. H&E staining results revealed that Model rats exhibited significant crypt atrophy, along with extensive immune cell infiltration in the colonic lamina propria. Both 5-ASA and SPS treatments demonstrated therapeutic efficacy by showing improved colonic mucosal architecture (Figure 2E). Ultrastructural analysis through TEM (Figure 2F) showed distinct pathological features in Model group: loss of intestinal microvilli, disrupted mitochondrial structure with markedly reduced mitochondrial numbers and increased swelling. Compared with the Model group, all treatment groups exhibited restored intestinal microvilli, increased mitochondrial quantity, repaired cristae structures, and reduced mitochondrial swelling. In conclusion, SPS significantly alleviated TNBS-induced clinical manifestations of colitis in rats, ameliorated colonic shortening, and promoted structural repair of colonic damage.

### 3.3. SPS Alleviates Colonic Inflammation and Modulates Systemic Immunity in Colitis Rats

QRT-PCR analysis revealed that the Model group exhibited a characteristic imbalance in inflammatory cytokines, with significantly upregulated mRNA expression of pro-inflammatory factors and decreased IL-10 expression. Following intervention with various concentrations of SPS, these inflammatory indicators showed dose-dependent improvement. Building upon previous findings from our research group demonstrating elevated CHI3L1 expression in DSS-induced colitis models, we conducted tracking experiments in TNBS-induced colitis rats. Both serum and colonic tissues showed significantly increased CHI3L1 expression, which was markedly downregulated after SPS treatment (Figure 3F–I). Additionally, MPO activity displayed significant upregulation in the Model group that was reduced by SPS intervention (Figure 3J).

In this study utilizing a TNBS-induced rat colitis model, comparative analysis revealed that the Model group exhibited characteristic imbalanced immune organ status with significantly decreased thymus index and markedly increased spleen index compared to the Control group. Notably, these abnormal parameters were substantially ameliorated following SPS intervention (Figure 3K,L). Further investigation of humoral immunity-related indicators demonstrated that the Model group showed significantly elevated serum IgG, IgM and sIgA levels compared to controls. Importantly, SPS treatment effectively downregulated these excessive immunoglobulin expressions (Figure 3M–O). These findings collectively confirm that SPS not only alleviated TNBS-induced colonic inflammation but also exerted immunomodulatory effects through regulating thymus/spleen indices and humoral immune responses.

### 3.4. SPS Regulates Macrophage M1 Polarization by Inhibiting the STAT3/NF-κB Pathway in Colitis Rats

To investigate the therapeutic mechanism of SPS in colitis, we examined the phosphorylation status of the STAT3/NF-κB signaling pathway. Our results demonstrated that the Model group exhibited significantly elevated phosphorylation levels of STAT3 and NF-κB, whereas these levels were markedly suppressed following SPS intervention (Figure 4A–C). These findings suggested that SPS may exert its anti-inflammatory effects by blocking the activation of the STAT3/NFκB pathway. Further analysis of macrophage polarization status in colonic tissues via flow cytometry revealed that the proportion of M1-type macrophages in the Model group was significantly increased compared to the Control group; however, H-SPS treatment markedly reversed this phenomenon (Figure 4D–H). This study demonstrated that SPS likely alleviates TNBS-induced colitis inflammation by suppressing the STAT3/NF-κB signaling pathway, thereby reducing M1 macrophage polarization.

### 3.5. The Anti-Inflammatory Effects of SPS on Caco-2 Cells

The aforementioned findings indicated that SPS significantly suppresses inflammatory responses, promoted structural repair of damaged epithelial cells, dynamically regulated immune homeostasis, and critically regulated macrophage polarization towards the M1 phenotype. Notably, CHI3L1 maybe serves as a key immunomodulatory molecule within this regulatory network. To validate these findings, an in vitro experimental system was established. Firstly, the anti-inflammatory effect of SPS was evaluated using a TNF-α-induced Caco-2 cell inflammation model. The results of cytotoxicity experiments showed that the cell survival rate remained above 80% in the concentration range of 10–200 ng/mL of TNF-α, indicating that the concentration range did not cause significant cytotoxicity (Figure 4A). Similarly, when the concentration of SPS was 10–320 μg/mL, the cell survival rate also remained above 80%, further confirming that the drug did not have significant cytotoxicity in this concentration range (Figure 4B). Through qRT-PCR screening, 25 ng/mL TNF-α induction for 24 h was identified as the optimal modeling concentration (Appendix A). Following co-incubation of Caco-2 cells with TNF-α and SPS (10–320 µg/mL), RT-PCR results revealed that SPS concentrations of 20–320 µg/mL effectively suppressed the expression of pro-inflammatory factors (Appendix A). Based on dose–response relationship analysis, three gradient concentrations of SPS (20, 40, and 80 μg/mL) were ultimately selected for subsequent experiments.

As illustrated in Figure 4, in the TNF-α-induced inflammatory model, the Model group exhibited a significant pro-inflammatory response compared to the Control group. The mRNA expression levels of pro-inflammatory cytokines IL-6, IL-1β, and TGF-β were markedly upregulated (*p* < 0.05), while the expression of the anti-inflammatory cytokine IL-10 was substantially suppressed, SPS intervention significantly reversed these inflammation-related gene expression abnormalities (Figure 5F). Furthermore, the Model group demonstrated a notable upregulation of CHI3L1 expression in Caco-2 cells, including increased intracellular CHI3L1 mRNA transcription levels, elevated intracellular protein content, and heightened secretory CHI3L1 protein levels in the culture medium, SPS treatment effectively attenuated this CHI3L1 overexpression phenotype (Figure 5G–J). Western blot validation of signaling pathways demonstrated that SPS significantly inhibited phosphorylation of STAT3 and NF-κB, with these findings closely aligned with the STAT3/NF-κB pathway suppression observed in animal experiments (Figure 5K–M).

### 3.6. The Regulation of SPS on LPS-Induced M1 Polarization in THP-1 Cells

Next, we further investigated the regulatory effects of SPS on LPS-induced M1 polarization in THP-1 cells. Cytotoxicity assessment revealed that cell viability remained >80% across all LPS treatment groups (10–200 ng/mL), confirming no significant cytotoxic effects within this concentration range (Appendix A). Through qRT-PCR screening, 100 ng/mL LPS induction for 24 h was identified as the optimal concentration for model establishment, three gradient concentrations of SPS (20, 40, and 80 μg/mL) were selected for subsequent experiments. The results demonstrated that LPS significantly induced M1 polarization of THP-1 cells, characterized by excessive production of nitric oxide (NO) and inducible nitric oxide synthase (iNOS), along with increased secretion of pro-inflammatory cytokines including tumor necrosis factor-α (TNF-α), interleukin-6 (IL-6), and interleukin-1β (IL-1β). Experimental data revealed that the expression levels of these inflammatory mediators were significantly up-regulated in the LPS-induced Model group (Figure 6A–E).

Notably, administration of SPS significantly downregulated the expression of the aforementioned inflammatory indicators. Further investigation revealed a marked upregulation of CHI3L1 expression in the Model group, which was effectively suppressed by SPS intervention (Figure 6F–I). In cellular phenotypic analysis, LPS induction triggered a marked increase in M1 macrophage surface markers CD80 and CD86, while SPS treatment significantly attenuated the elevated levels of these polarization markers, demonstrating its inhibitory effect on M1 polarization (Figure 6M–P). Mechanistic studies indicated that SPS likely modulates inflammatory responses by inhibiting STAT3 and NF-κB phosphorylation, thereby reversing LPS-induced M1 polarization and downregulating CHI3L1 expression (Figure 6J–L). Collectively, these findings demonstrate that SPS suppresses inflammatory responses and regulates macrophage M1 polarization through modulation of the STAT3/NF-κB signaling pathway.

### 3.7. Effect of Inflammatory Damage in Caco-2 Cells on M1 Polarization of Macrophages

Previous studies have demonstrated that SPS significantly ameliorates LPS-induced inflammatory injury in THP-1 macrophages and TNF-α-triggered inflammatory damage in Caco-2 intestinal epithelial cells by suppressing the activation of NF-κB/STAT3 signaling pathway, while regulating macrophage M1 polarization, with CHI3L1 identified as a key participant in this process. Building on this evidence, the current study hypothesized that epithelial-immune cell crosstalk may exist in colitis, where inflammatory-injured epithelial cells promote macrophage polarization toward pro-inflammatory M1 phenotype through excessive CHI3L1 secretion. To validate this hypothesis, we established an epithelial cell-macrophage co-culture system where TNF-α-induced inflammatory Caco-2 cells were co-cultured with THP-1-derived macrophages. QRT-PCR analysis revealed that compared to the TNF-α stimulation alone group, the co-culture system exhibited significantly reduced mRNA expression levels of inflammatory mediators including NO, TNF-α, IL-6, IL-1β, and iNOS (Figure 7A–E). Notably, THP-1 cells co-cultured with inflamed Caco-2 cells demonstrated distinct M1 polarization characteristics accompanied by upregulated CHI3L1 expression; however, SPS intervention substantially suppressed this M1 polarization phenomenon and restored CHI3L1 expression to near-normal levels (Figure 7F–M).

### 3.8. The Role of CHI3L1 in M1 Macrophage Polarization Induced by Inflammatory-Damaged Caco-2 Cells

To investigate whether CHI3L1 mediates M1 macrophage polarization induced by inflammatory injury in Caco-2 cells, this study employed two intervention approaches: gene silencing (CHI3L1-siRNA) and recombinant protein overexpression (r-CHI3L1). The knockdown and overexpression efficiencies were validated using qRT-PCR and Western blot (Appendix A). Ultimately, the overexpression model was selected as the exogenous addition of 100 ng/mL CHI3L1, while the knockdown model employed an siRNA sequence with the designation CHI3L1-Homo-130. Experimental data revealed that CHI3L1 overexpression significantly exacerbated the inflammatory response triggered by epithelial cell injury, while SPS treatment partially reversed this pro-inflammatory effect. Conversely, CHI3L1 knockdown not only reduced baseline inflammation levels but also exhibited synergistic anti-inflammatory effects when combined with SPS (Figure 8A–E).

QRT-PCR and flow cytometry further confirmed a positive correlation between CHI3L1 expression levels and the degree of M1 macrophage polarization (Figure 8F–I). That is to say, overexpression of CHI3L1 in epithelial cells increased the proportion of M1 polarization in immune cells and promoted the secretion of pro-inflammatory factors such as IL-6, TGF-β and TNF-α. Mechanistic studies demonstrated that CHI3L1 regulates macrophage polarization by modulating the phosphorylation status of the STAT3/NF-κB signaling pathway—knockdown of CHI3L1 inhibited STAT3/NF-κB phosphorylation, whereas overexpression activated this pathway.

## 4. Discussion

In recent years, due to their low toxicity and excellent pharmacological activities such as anti-inflammatory [26], anti-tumor [27], immunomodulatory [28], and antioxidant effects, Traditional Chinese medicine polysaccharides have become a hotspot in the development of functional food. Previous studies have shown that SPSs are one of the primary bioactive components of safflower, mainly composed of monosaccharides including Man, Glu, Gal, Rha, Ara, and Xyl [29,30]. They exhibit various biological effects such as free radical scavenging, antioxidant activity [9], inhibition of colon cancer, and suppression of tongue squamous cell carcinoma. This study revealed that SPS is primarily composed of Rha, Ara, Gal, and Glu, with an average molecular weight of 31.11 kDa.

Colitis is an immune-mediated inflammatory bowel disease that originates from inflammation of the rectal mucosa and extends proximally to the colon. Currently, treatment options for Colitis are limited and mainly include fecal microbiota transplantation (FMT), hyperbaric oxygen therapy, aminosalicylates, immunosuppressants, and surgical interventions. However, these therapies are associated with adverse effects and poor prognoses, yielding suboptimal therapeutic outcomes. In recent years, growing research has demonstrated the promising potential of herbal polysaccharides in treating colitis [31,32]. For instance, Polygonatum sibiricum polysaccharides alleviated ulcerative colitis by modulating inflammatory immune responses independently of the gut microbiota [33], while Phellinus linteus polysaccharides mitigated DSS-induced colitis by reducing inflammation and oxidative stress, regulating gut microbiota composition, and restoring normal intestinal barrier function [34]. Our previous study confirmed that SPS ameliorates colitis by restoring the intestinal mucosal barrier and modulating gut microbiota [10]. However, whether SPS treats colitis through immune regulation remains unclear. In this study, we employed both in vivo and in vitro experiments to investigate the therapeutic efficacy and immune-related mechanisms of SPS in colitis treatment.

First, we established an ethanol/TNBS-induced colitis model in rats [35]. Ethanol can disrupt the colonic barrier, allowing TNBS to penetrate into the submucosal lamina propria and induce transmural colitis along with a Th1-mediated immune response. In our experiments, we evaluated the therapeutic effects of SPS on TNBS-induced colitis. Rats with TNBS-induced colitis exhibited reduced food and water intake, weight loss, decreased DAI scores, shortened colon length, and pathological changes in colon tissue. However, after administration of different doses of SPS, significant improvements were observed in food/water intake, weight loss, colon length, DAI scores, and histological scores. These findings demonstrate that SPS effectively alleviates TNBS-induced clinical symptoms of colitis, ameliorates colon shortening, repairs colonic mucosal damage, maintains epithelial barrier function, and reduces inflammatory infiltration.

Growing evidence indicates that during the onset of colitis, the body produces large amounts of pro-inflammatory cytokines (IL-6, TNF-α, IL-1β, etc.), primarily derived from macrophages and neutrophils. The excessive expression of these pro-inflammatory cytokines further amplifies the inflammatory response, exacerbates intestinal tissue damage [36], and mediates immune reactions. In our experiments, TNBS induction elevated the mRNA levels of IL-6, IL-1β, and TNF-α in rat colon tissues, while SPS reduced their expression, exhibiting effects similar to those of 5-ASA. Additionally, SPS increased the expression of the anti-inflammatory cytokine IL-10. These results demonstrate that SPS can regulate the imbalance between pro-inflammatory and anti-inflammatory cytokine expression. This phenomenon has also been confirmed in intestinal epithelial cell models and M1-polarized macrophages.

The excessive activation of immune cells is a core process in the development of colitis. It has been found that the predominant immunoglobulin on mucosal surfaces is sIgA [37], while IgG is the most abundant antibody in serum [38], and IgM has been shown to potentially play a protective role in autoimmunity [39]—all of which are key components of humoral and mucosal immunity. Macrophages are critical immune cells involved in Colitis, with IFN-γ, TNF-α, and microbial products like LPS stimulating their M1 polarization. TLR2, TLR4, CD80, and CD86 are surface receptors expressed on M1 macrophages. Additionally, we evaluated the expression of MPO in the colonic tissues of colitis rats. MPO is a heme protease specifically secreted by neutrophils and serves as an important biochemical marker of inflammation. It has been reported that almond polysaccharides can reduce MPO activity and pro-inflammatory factor levels in DSS-induced Colitis mice [31]. Our findings indicate that SPS can lower the expression levels of IgG and IgM while downregulating MPO expression, suggesting that SPS may mediate immune regulation to alleviate ulcerative colitis.

Macrophages are a crucial type of immune cells in animals, abundantly present in the intestinal mucosa and playing a pivotal role in regulating intestinal homeostasis. In the context of colitis, a large number of macrophages accumulate in the lamina propria near the epithelium and overexpress pro-inflammatory cytokines such as IL-1β and TNF-α [40], indicating macrophage polarization. Colitis patients exhibit a higher proportion of M1 macrophages in their tissues [41], and this imbalance in macrophage polarization significantly impacts colitis progression. Consequently, an increasing number of studies aim to alleviate or treat colitis by modulating macrophage polarization states. Research has shown that macrophages isolated from TNBS-induced colitis mice display M1 polarization [42], which may activate signaling pathways such as NF-κB, JAK/STAT, and ERK, releasing large amounts of pro-inflammatory cytokines (TNF-α, IL-1β, IL-23) and chemokines (CXCL8, CCL2) to participate in immune responses. Our study demonstrated that TNBS-induced colitis rats exhibit an elevated proportion of M1-polarized macrophages, while SPS can reduce the M1 polarization level of intestinal macrophages. In the M1-polarized macrophage model, it was also found that SPS could inhibit the M1 polarization process.

NF-κB is a crucial intracellular signaling regulator that modulates a vast array of cytokines involved in diverse immune and inflammatory processes. Recognized as a key modulator of inflammatory responses, NF-κB not only mediates the induction of various pro-inflammatory genes in innate immune cells but also regulates the activation, differentiation, and effector functions of inflammatory T cells. Previous studies have demonstrated that nervonic acid ameliorates colitis by inhibiting the NF-κB signaling pathway in DSS-induced colitis mice [42]. STAT3, a member of the signal transducer and activator of transcription (STAT) family, participates in multiple physiological activities, including inflammation and immunity. Oleanolic acid can directly bind to SHP2, and inhibit STAT3 phosphorylation and Th17 differentiation, thereby alleviating inflammatory responses and mitigating experimental colitis induced by TNBS in mice [43]. Research has shown that STAT3 remains inactive in unstimulated cells but becomes activated upon stimulation by cytokines (e.g., IL-6, TNF-α) and growth factors, undergoing phosphorylation at Tyr705 and Ser727 residues to form dimers that translocate into the nucleus [44,45]. Both our in vivo and in vitro experiments demonstrated that SPS may alleviate colitis inflammation and modulate the intestinal immune system through the STAT3/NF-κB signaling pathway.

Studies have shown that the expression of CHI3L1 is significantly upregulated in various inflammatory diseases, including chronic enteritis, pneumonia, asthma, and arthritis. It promoted the release of pro-inflammatory factors by activating signaling pathways such as NF-κB, exacerbating the deterioration of the tissue inflammatory microenvironment. Under chronic inflammatory conditions, CHI3L1 activates downstream inflammatory and carcinogenic pathways through apical expression in epithelial cells and is closely associated with macrophage polarization. It induces M2 macrophage differentiation, suppresses innate immune responses, and simultaneously stimulates immune checkpoint molecules, creating a vicious cycle of pro-inflammation and immune suppression. Additionally, previous research results demonstrated that CHI3L1 was specifically upregulated in the colonic epithelial cells and lamina propria macrophages of experimental murine colitis [45]. In our study, CHI3L1 expression was significantly elevated in colitis rats as well as in epithelial cells and macrophages, which aligns with prior findings.

In this study, we revealed that SPS enhances immune function and alleviates intestinal inflammation by inhibiting the STAT3/NF-κB signaling pathway, suppressing M1 macrophage polarization, and reducing CHI3L1 expression. Through Transwell experiments, we confirmed that CHI3L1 mediates inflammation-induced M1 macrophage polarization in epithelial cells. Furthermore, knockdown and overexpression experiments demonstrated that CHI3L1 knockdown partially inhibits macrophage polarization, which may be regulated via the STAT3/NF-κB pathway. In conclusion, CHI3L1 serves as a key molecule linking epithelial inflammation to macrophage polarization, and its mechanism of regulating the immune microenvironment through the STAT3/NF-κB pathway provides a novel therapeutic target for colitis. Meanwhile, SPS exhibits significant therapeutic potential for colitis through multi-target interventions. In view of the clear biological activity of SPS in alleviating intestinal inflammation, regulating immune-related pathways and targeting CHI3L1 in the treatment of Colitis, it can provide a direct and reliable scientific basis for the subsequent development of functional foods using SPSs as a core functional component.

## Figures and Tables

**Figure 1 foods-14-03199-f001:**
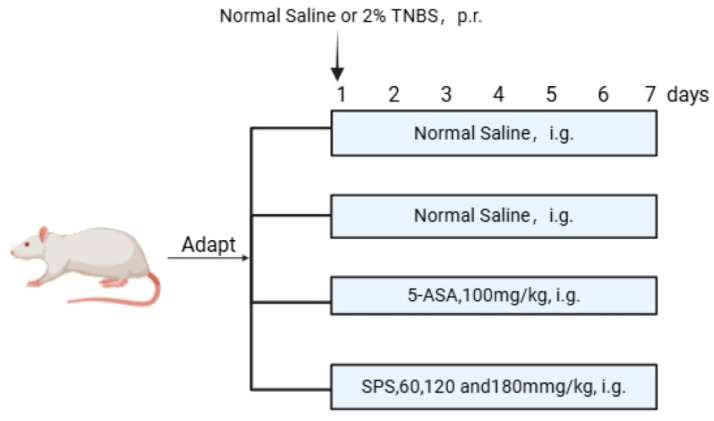
Schematic illustration of rat colitis injury model construction.

**Figure 2 foods-14-03199-f002:**
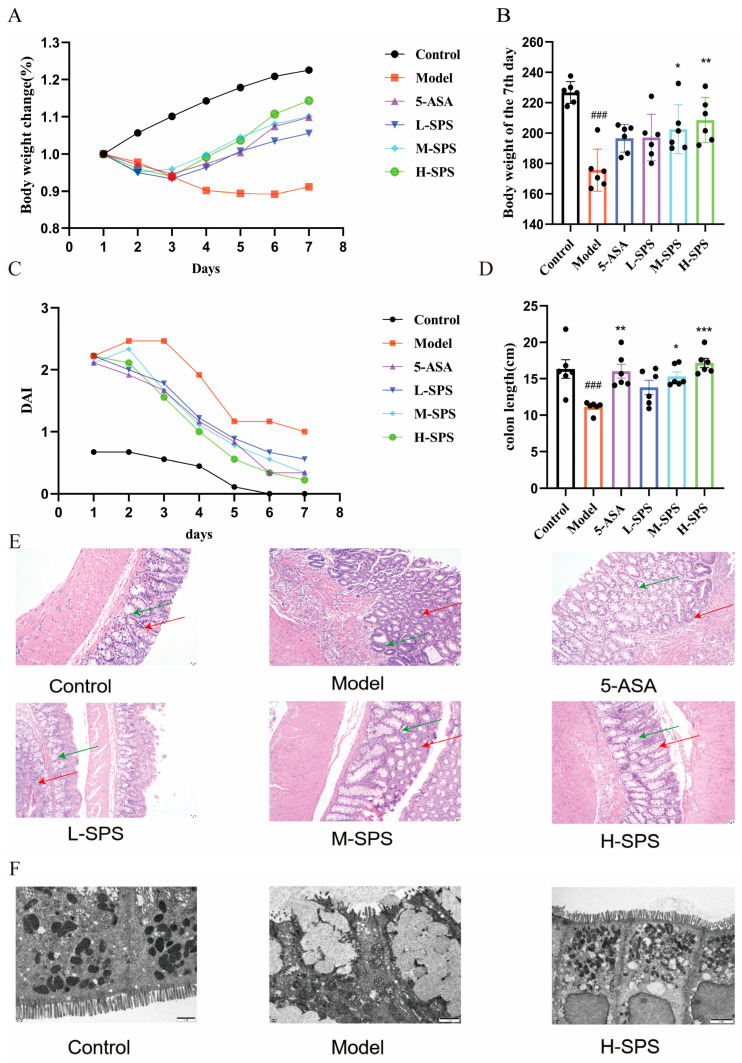
SPS alleviates symptoms of TNBS-induced colitis and reduces colonic damage in rats. (**A**) The body weight changes in rats. (**B**) Body weight of the 7th day. (**C**) DAI score. (**D**) Colon length. (**E**) The H&E staining of colon (red arrow represent inflammatory cell infiltration, green arrows represent crypt integrity destruction, 20×). (**F**) The TEM imaging of colon (scale bars 1  µm). The data are presented as the mean ± SEM, *n* ≥ 3. ^###^ *p* < 0.001 versus the Control group. * *p* < 0.05, ** *p* < 0.01 and *** *p* < 0.001 versus the Model group.

**Figure 3 foods-14-03199-f003:**
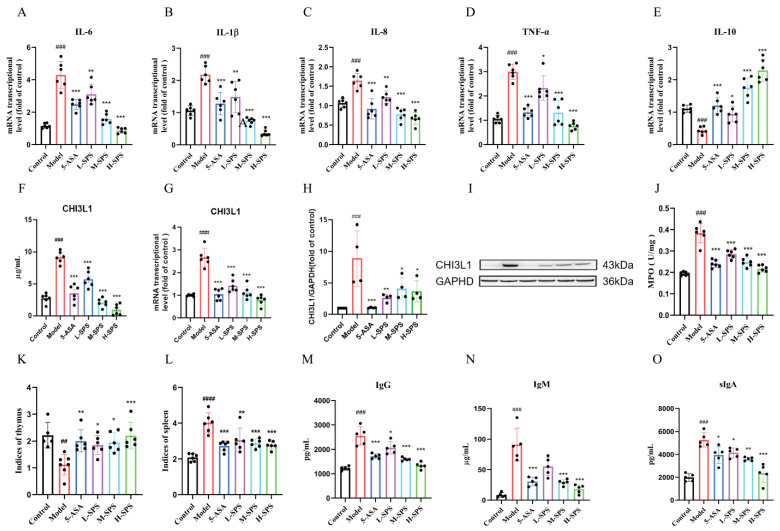
Effects of SPS on colonic inflammation and systemic immunity in colitis rats. The mRNA levels of interleukin-6 (**A**), interleukin-1β (**B**), interleukin-8 (**C**), tumor necrosis factor-α (**D**), interleukin-10 (**E**), CHI3L1. (**G**) the relative quantification was determined using the 2^−ΔΔCt^ method normalized to β-actin. (**F**) CHI3L1 concentration in serum. (**H**) CHI3L1 relative protein expression statistics. (**I**) Protein expression level of CHI3L1. (**J**) The MPO activity. (**K**) Indices of thymus. (**L**) Indices of spleen. (**M**) IgG concentration in serum. (**N**) IgM concentration in serum. (**O**) sIgA concentration in colons. The data are presented as the mean ± SEM, *n* ≥ 5. ^##^ *p* < 0.01 versus the Control group. ^###^ *p* < 0.001 versus the Control group. * *p* < 0.05, ** *p* < 0.01 and *** *p* < 0.001 versus the Model group.

**Figure 4 foods-14-03199-f004:**
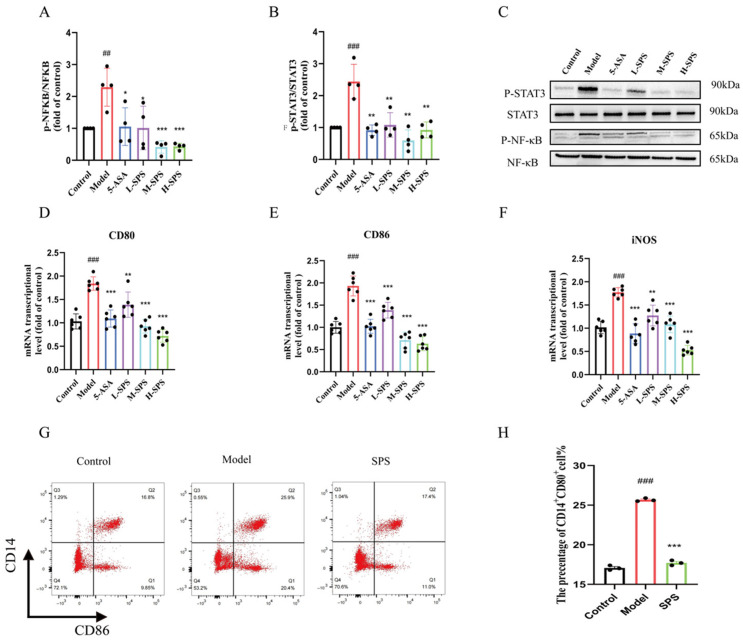
SPS regulates M1 macrophage polarization via the STAT3/NF-κB pathway. (**A**) Quantitative assessment of p-NF-κB protein levels. (**B**) Quantitative assessment of p-STAT3 protein levels. (**C**) The protein expression levels of p-STAT3 and p-NF-κB. (**D**–**F**) The mRNA expression levels of CD80, CD86 and iNOS. (**G**) Cell ratio of CD14+CD80+ in cells in detected by flow cytometry. (**H**) The change in cell ratio of CD14+CD80+ expression. The data are presented as the mean ± SEM, n ≥ 3. ^##^ *p* < 0.01 and ^###^ *p* < 0.001 versus the Control group. * *p* < 0.05, ** *p* < 0.01 and *** *p* < 0.001 versus the Model group.

**Figure 5 foods-14-03199-f005:**
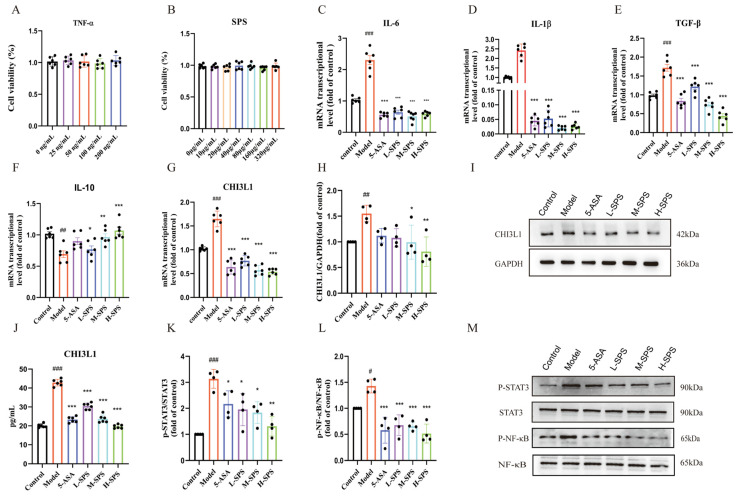
SPS alleviates TNF-α-induced inflammatory injury in epithelial cells. (**A**) The cytotoxic impact of TNF-α on Caco-2 cells. (**B**) The cytotoxic impact of SPS on Caco-2 cells. (**C**–**G**) The mRNA expression levels of TNF-α, IL-1β, TGF-β, IL-10, CHI3L1. (**J**) CHI3L1 concentration in Cell culture medium. (**H**) CHI3L1 relative protein expression statistics. (**I**) Protein expression level of CHI3L1. (**K**) The relative expression of p-STAT3 protein statistics. (**L**) The relative expression of p-NF-κB protein statistics. (**M**) NF-κB p65, p-NF-κBp65, STAT3 and p-STAT3 protein expression levels. The data are presented as the mean ± SEM, *n* ≥ 4. ^#^ *p* < 0.05, ^##^ *p* < 0.01 and ^###^ *p* < 0.001 versus the Control group. * *p* < 0.05, ** *p* < 0.01 and *** *p* < 0.001 versus the Model group.

**Figure 6 foods-14-03199-f006:**
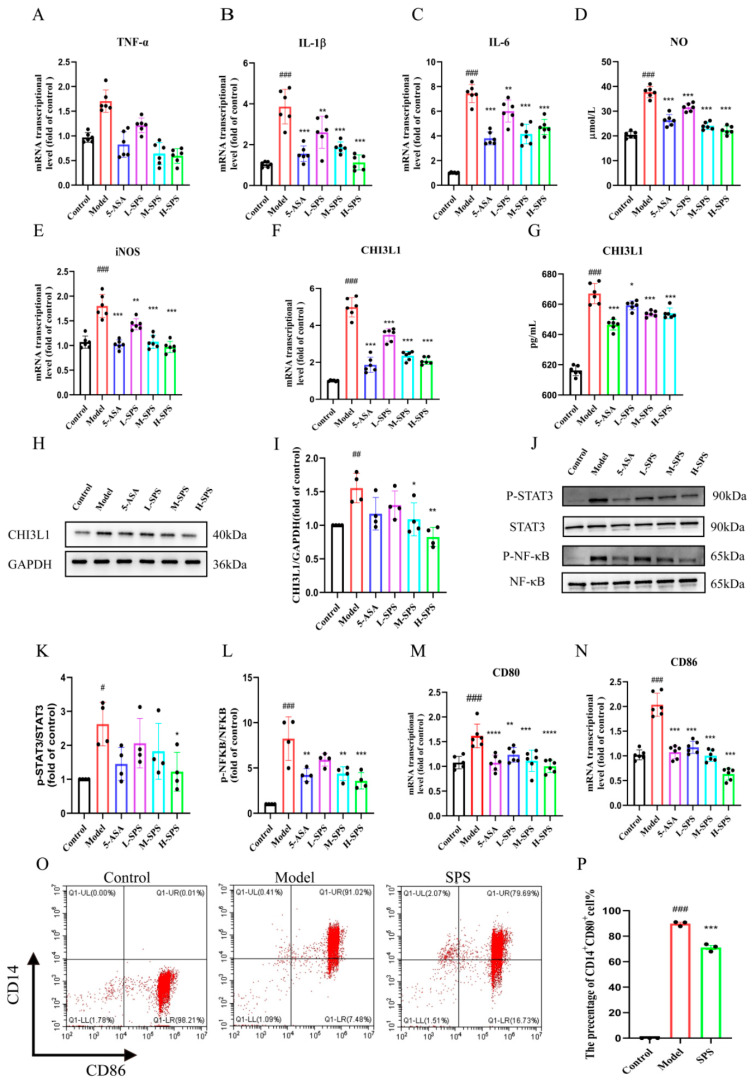
SPS alleviates LPS-induced M1 macrophage polarization. (**A**–**C**) The mRNA levels of TNF-α, IL-6, IL-1β, iNOS. (**D**) NO concentration in colons. (**E**,**F**) The mRNA levels of iNOS, CHI3L1. (**G**) CHI3L1 concentration in Cell culture medium. (**H**) Protein expression level of CHI3L1. (**I**) CHI3L1 relative protein expression statistics. (**J**) NF-κB p65, p-NF-κBp65, STAT3 and p-STAT3 proteinexpression levels. (**K**,**L**) The relative expression of p-NF-κB p65/p-STAT3 protein statistics. (**M**,**N**) The mRNA expression levels of CD80, CD86. (**O**) Cell ratio of CD14+CD80+ in cells in detected by flow cytometry. (**P**) The change in cell ratio of CD14+CD80+ expression. Results are expressed as mean ± SEM, *n* ≥ 3. ^#^ *p* < 0.05, ^##^ *p* < 0.01 and ^###^ *p* < 0.001 versus the Control group. * *p* < 0.05, ** *p* < 0.01 and *** *p* < 0.001 versus the Model group.

**Figure 7 foods-14-03199-f007:**
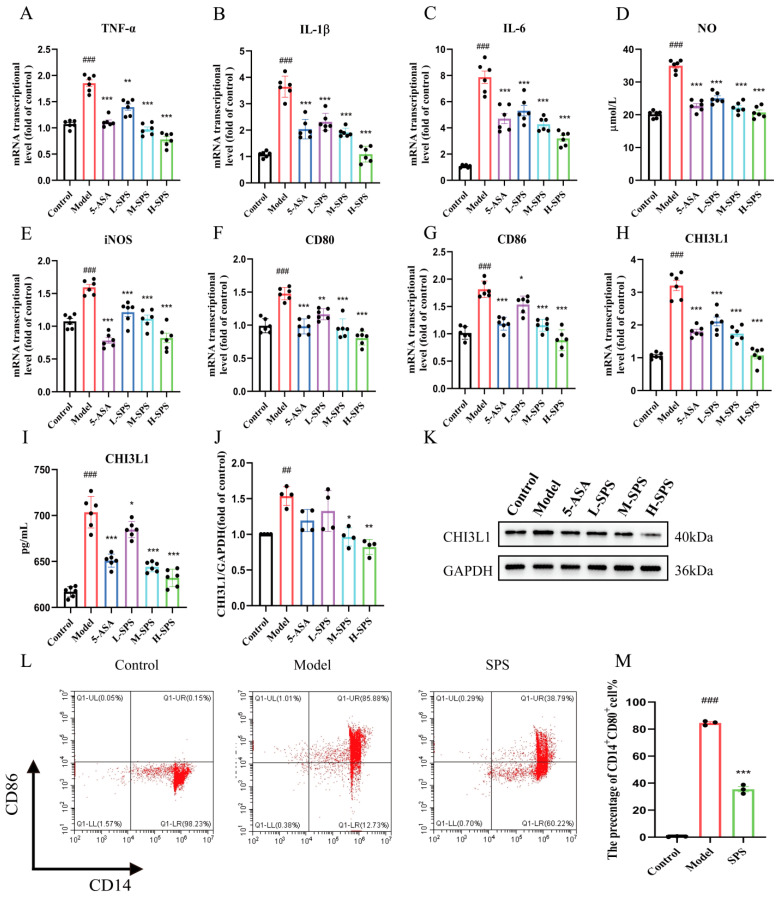
The impact of inflammatory-damaged Caco-2 cells on macrophage M1 polarization and the role of SPS. (**A**–**C**) The mRNA levels of TNF-α,IL-6, IL-1β. (**D**) NO concentration in colons. (**E**–**H**) The mRNA levels of iNOS,CD80,CD86 andCHI3L1. (**I**) CHI3L1 concentration in cell culture medium. (**J**) Protein expression level of CHI3L1. (**K**) CHI3L1 relative protein expression statistics. (**L**) Cell ratio of CD14+CD80+ in cells in detected by flow cytometry. (**M**) The change in cell ratio of CD14+CD80+ expression. Results are expressed as mean ± SEM, *n* ≥ 3. Statistical significance is denoted as: ^##^ *p* < 0.01 and ^###^ *p* < 0.001 versus the Control group. * *p* < 0.05, ** *p* < 0.01 and *** *p* < 0.001 versus the Model group.

**Figure 8 foods-14-03199-f008:**
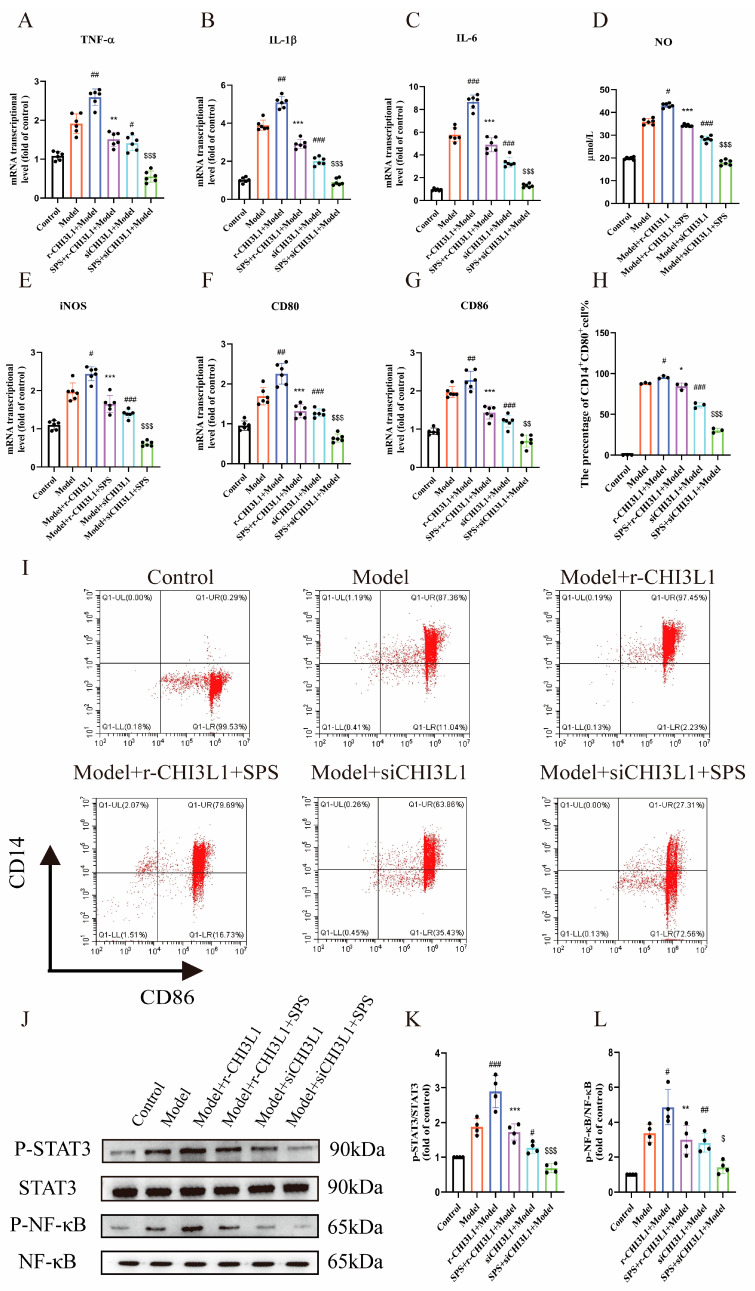
The role of CHI3L1 in epithelial-immune cell interactions and the function of SPS. (**A**–**C**) The mRNA levels of TNF-α,IL-6, IL-1β. (**D**) NO concentration in colons. (**E**–**G**) The mRNA levels of iNOS, CD80 andCD86. (**H**) The change in cell ratio of CD14+CD80+ expression. (**I**) Cell ratio of CD14+CD80+ in cells in detected by flow cytometry. (**J**) NF-κB p65, p-NF-κBp65, STAT3 and p-STAT3 proteinexpression levels. (**K**,**L**) NF-κB p65, p-NF-κBp65, STAT3 and p-STAT3 protein expression levels. The data are presented as the mean ± SEM, n ≥ 3. ^#^
*p* < 0.05, ^##^ *p* < 0.01 and ^###^ *p* < 0.001 versus the Model group. * *p* < 0.05, ** *p* < 0.01 and *** *p* < 0.001 versus the r-CHI3L1+SPS group, ^$^ *p* < 0.05, ^$$^ *p* < 0.01 and ^$$$^
*p* < 0.001 versus the si-CHI3L1+SPS group.

**Table 1 foods-14-03199-t001:** Scoring system for disease activity index (DAI) in the rats.

Score	Loss of Body Weight (%)	Loose Stools	Bloody Stools
0	<1	olid and granular	negative expression of occult blood
1	1–5	soft and granular	weakly positive expression of occult blood
2	5–10	semi formed loose stool	positive expression of occult blood
3	10–15	unformed loose stool	strongly positive expression of occult blood
4	≥15	had sign of liquid	strongly positive and visible blood

**Table 2 foods-14-03199-t002:** Information about primers.

Gene	Forward Primer (5′–3′)	Reverse Primer (5′–3′)
IL-6(R)	TACCACTTCACAAGTCGGAGGC	CTGCAAGTGCATCATCGTTGTTC
IL-10(R)	CGGGAAGACAATAACTGCACCC	CGGTTAGCAGTATGTTGTCCAGC
IL-1β(R)	TTCGACACATGGGATAACGAGG	TTTTTGCTGTGAGTCCCGGAG
iNOS(R)	GGAGGACCACCTCTATCAGG	CCTGAACGTAGACCTTGGGT
CD80(R)	CCTCAAGTTTCCATGTCCAAGGC	GAGGAGAGTTGTAACGGCAAGG
CD86(R)	ACGTATTGGAAGGAGATTACAGCT	TCTGTCAGCGTTACTATCCCGC
TNF-α(R)	CCCTCACACTCAGATCATCTTCT	GCTACGACGTGGGCTACAG
CHI3L1(R)	CCACAGTCCATAGAATCCTCGG	TGCCTGTCCTTCAGGTACTGCA
IL-8(R)	CAGGCTTCCTTGTGCAAGTG	TCGAAAGCTGCTATTTCACAG
GAPDH	CATCACTGCCACCCAGAAGACTG	ATGCCAGTGAGCTTCCCGTTCAG
IL-6(H)	TAGTCCTTCCTACCCCAATTTCC	TTGGTCCTTAGCCACTCCTTC
IL-10(H)	GCTCTTACTGACTGGCATGAG	CGCAGCTCTAGGAGCATGTG
IL-1β(H)	CTCAACTGTGAAATGCCACC	GAGTGATACTGCCTGCCTGA
iNOS(H)	CAGAAGTCAAAGTCTCAGACAT	GTCATCTTGTATTGTTGGGCT
CD80(H)	CCTCAAGTTTCCATGTCCAAGGC	GAGGAGAGTTGTAACGGCAAGG
CD86(H)	ACGTATTGGAAGGAGATTACAGCT	TCTGTCAGCGTTACTATCCCGC
TNF-α(H)	CCGAGTCTGGGCAGGTCTA	GTCTGAAGGAGGGGGTAAT
CHI3L1(H)	CCACAGTCCATAGAATCCTCGG	TGCCTGTCCTTCAGGTACTGCA
TGF-β(H)	CAGGCTTCCTTGTGCAAGTG	TCGAAAGCTGCTATTTCACAG

## Data Availability

The original contributions presented in the study are included in the article/Appendix A, further inquiries can be directed to the corresponding authors.

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
