# Peer review of "Safflower Polysaccharides Alleviate TNBS-Induced Colitis by Modulating Gut Immunity"

_foods, 2025, doi:10.3390/foods14183199_

Round 1
Reviewer 1 Report
Comments and Suggestions for Authors
Dear editor,
The manuscript “Safflower polysaccharides alleviate ulcerative colitis by modu-1 lating gut immunity” by Jiang et al. deals with the immunological mechanisms of these polysaccharides an animal model of ulcerative colitis.
The manuscript is well developed, and the biological experiments section is noteworthy, but still has flaws regarding the structural characterization experiments needed in order to determine the structural features of the polysaccharide. Particularly NMR and methylation data are missing.
I would not recommend this manuscript in the state that it is now, however if the authors perform NMR and methylation analysis in order to characterized the fine structural features of the polysaccharide and perform the reviews listed below it would make the manuscript suitable for a resubmission.
Minor revisions:
Page 1, Line 3: Please standardize the author names font size.
Page 1, Line 8: “This study aims” or “The present study aims”
Page 1, Line 11: There are spaces missing. Same on line 14, 16 and 17.
Page 2, Line 61: There are spaces missing.
Page 3, Line 77: There are spaces missing.
Page 3, Line 81: There are spaces missing.
Page 3, Line 97: There are spaces missing.
Page 3, Line 98: There are extra spaces.
Page 4, Line 129: The chemical formulas should be written with subscript numbers. Also there is a space missing before the M symbol.
Page 4, Line 154: Number of the ethical committee approval document?
Page 7, Lines 228 and 229: There are spaces missing.
Page 8, Line 245: It appears that there are 2 font types, I suggest the authors to standardize the font type throughout the entire length of the manuscript.
Author Response
1. The manuscript is well developed, and the biological experiments section is noteworthy, but still has flaws regarding the structural characterization experiments needed in order to determine the structural features of the polysaccharide. Particularly NMR and methylation data are missing.
Response: Thank you very much for your comments.We acknowledge that this aspect was indeed not considered in our study, representing one of the limitations of the current experiment. Additionally, due to time constraints, we were unable to complete supplementary experiments addressing this issue within the given timeframe. However, our research group will implement comprehensive follow-up studies to further investigate and refine this aspect. The reviewer's suggestion of the NMR and methylation data has been studied as a separate problem by another team member, and the experimental results are expected to be published in the near future.
2. Minor revisions:
Page 1, Line 3: Please standardize the author names font size.
Page 1, Line 8: “This study aims” or “The present study aims”
Page 1, Line 11: There are spaces missing. Same on line 14, 16 and 17.
Page 2, Line 61: There are spaces missing.
Page 3, Line 77: There are spaces missing.
Page 3, Line 81: There are spaces missing.
Page 3, Line 97: There are spaces missing.
Page 3, Line 98: There are extra spaces.
Page 4, Line 129: The chemical formulas should be written with subscript numbers. Also there is a space missing before the M symbol.
Page 4, Line 154: Number of the ethical committee approval document?
Page 7, Lines 228 and 229: There are spaces missing.
Page 8, Line 245: It appears that there are 2 font types, I suggest the authors to standardize the font type throughout the entire length of the manuscript.
Response: Thanks for your comments, and We have modified the spaces of the whole article according to your suggestion. We have standardized the font size of all author names in the manuscript in accordance with the journal's formatting requirements(page 1, line 3),all chemical formulas have been adjusted to use subscript numbers for standardization (page 4, line 143). Also we have supplemented the animal ethics statement and placed it in Section 2.4 (page 4, line 173).
Reviewer 2 Report
Comments and Suggestions for Authors
In this original article, Jiang et al. explored the protective effects of safflower polysaccharides (SPS) in a 2,4,6-trinitrobenzene sulphonic acid (TNBS)-induced colitis model in Sprague-Dawley rats. They also investigated the protective effects of SPS in Caco-2 cells, THP-1 cells, and also in a co-culture cell model. Their findings indicate that SPS has anti-inflammatory effects, and the underlying mechanisms seem to be associated with STAT3/NF-κB signalling and immunomodulation.
The MS is interesting and the experiments are well-designed; however, I have the following concerns and suggestions to improve the MS:
- I suggest reviewing the MS by a native speaker or a professional editing service, as some sentences are difficult to understand due to incorrect word order and grammatical errors. In addition, the MS contains multiple spacing inconsistencies (missing or extra spaces).
- The Authors refer to the TNBS model as a reliable model for ulcerative colitis, but this is incorrect. TNBS induces a phenotype similar to Crohn’s disease, characterised by a Th1/Th17 immune response and transmural inflammation. Please correct this issue in the MS and use "TNBS-induced colitis" instead. Furthermore, in the discussion lines 477-481, there is misleading information, such as the following:
- "transmural ulcerative colitis": TNBS indeed induce a transmural inflammation; however, ulcerative colitis is characterised by a superficial ulceration of the colon.
- "closely mimicking the clinical manifestations of human ulcerative colitis": this is also an incorrect statement, as TNBS-induced colitis models Crohn’s disease rather than ulcerative colitis.
- Also, I suggest including only "TNBS-induced colitis" in the title and not ulcerative colitis.
- While Supplementary Figure 1 provides this information, the Methods section (2.4) lacks several key experimental details (including the TNBS dose and vehicle, the SPS administration route, and the SPS doses in mg/kg), which are essential for ensuring the study’s reproducibility. Furthermore, it would be more appropriate to include the protocol figure in the main MS rather than in the supplementary section.
- Please use the passive voice where possible, instead of the current active voice in the Methods section. The verb tense is appropriate up to section 2.5; however, sections 2.6, 2.7, 2.9, 2.10, and 2.11 are not written in the correct tense. Kindly revise these accordingly.
- If two types of statistical tests were used (ANOVA and t-test), please indicate which test was applied in each case. Additionally, specify which post hoc test was used in the case of ANOVA.
- Please review the figure legends, as the panel orders are incorrect in several figures, such as Figure 3, Figure 4, Figure 5, Figure 6, and Figure 7. Kindly correct them.
- In the case of Figure 7, a new type of statistical significance annotation appears, which differs from the previous figures and is not consistent with the figure legends. I recommend using the same scheme for Figure 7 as in the other figures throughout the MS.
- Please provide information on what type of kit was used for the measurement of MPO activity. Also, in line 292, the Authors state that "the expression of MPO activity". Please clarify if the activity or expression was measured.
- Please add visible scale bars to the H&E and TEM images. It would be beneficial if the specific alterations were marked on the H&E-stained images with arrows (distortion of crypt architecture, thickening of the muscularis mucosae, etc.). Please also include the magnification in the figure legend.
- While analysing the NF-κB pathway, did the authors assess whether IκB expression or phosphorylation was affected?
- I suggest reconsidering the summary figure, as it is currently difficult to interpret and contains too much data. Instead of presenting individual results, it would be more effective to summarise the key conclusions drawn from the findings.
Author Response
1. I suggest reviewing the MS by a native speaker or a professional editing service, as some sentences are difficult to understand due to incorrect word order and grammatical errors. In addition, the MS contains multiple spacing inconsistencies (missing or extra spaces).
Response: Thank you very much for pointing out the language and formatting issues ! We have carried out a comprehensive review of the manuscript, corrected grammatical errors and space inconsistencies and other issues, to ensure that the language is clear and accurate, standardized and unified format. And here we did not list the changes but marked in red in the revised paper.
2.The Authors refer to the TNBS model as a reliable model for ulcerative colitis, but this is incorrect. TNBS induces a phenotype similar to Crohn’s disease, characterised by a Th1/Th17 immune response and transmural inflammation. Please correct this issue in the MS and use "TNBS-induced colitis" instead. Furthermore, in the discussion lines 477-481, there is misleading information, such as the following:
• "transmural ulcerative colitis": TNBS indeed induce a transmural inflammation; however, ulcerative colitis is characterised by a superficial ulceration of the colon.
• "closely mimicking the clinical manifestations of human ulcerative colitis": this is also an incorrect statement, as TNBS-induced colitis models Crohn’s disease rather than ulcerative colitis.
Response: Thank you very much for your valuable comments.We have changed TNBS-induced ulcerative colitis to "TNBS-induced colitis" in this MS (page 1, line 1).
We have deleted "ulcerative" from the phrase "Ethanol can disrupt the colonic barrier, allowing TNBS to penetrate into the submucosal lamina propria and induce transmural ulcerative colitis along with a Th1-mediated immune response"(page 20, line 519).
Finally, we deleted the passage "This model is characterized by infiltration of CD4 + cells, neutrophils, and macrophages into the lamina propria, accompanied by cytokine secretion [ 36 ], closely mimic colitisking the clinical manifestations of human ulcerative [ 37 ]"(page 20, line 521).
3. Also, I suggest including only "TNBS-induced colitis" in the title and not ulcerative colitis.
Response: Thank you very much for your valuable comments. We have changed "ulcerative colitis" to "TNBS-induced colitis" in the title (page 1, line 1).
4. While Supplementary Figure 1 provides this information, the Methods section (2.4) lacks several key experimental details (including the TNBS dose and vehicle, the SPS administration route, and the SPS doses in mg/kg), which are essential for ensuring the study’s reproducibility. Furthermore, it would be more appropriate to include the protocol figure in the main MS rather than in the supplementary section.
Response: Thanks for your advice.I have supplemented details such as TNBS dose and vehicle, the SPS administration route, and the SPS doses in mg / kg in the manuscript. In addition, I have put the experimental scheme diagram in the main paper (page 4, line 164-169).
5. Please use the passive voice where possible, instead of the current active voice in the Methods section. The verb tense is appropriate up to section 2.5; however, sections 2.6, 2.7, 2.9, 2.10, and 2.11 are not written in the correct tense. Kindly revise these accordingly.
Response: Thanks for your advice. The tenses of 2.6, 2.7, 2.9, 2.10 and 2.11 have been changed into passive voice in the manuscript (page 5, line 185-247).
6. 6.If two types of statistical tests were used (ANOVA and t-test), please indicate which test was applied in each case. Additionally, specify which post hoc test was used in the case of ANOVA.
Response: Thanks for your advice. With t-tests used to compare between-group data and one-way ANOVA for multiple-group comparisons in this MS (page 5, line 251-252).
7. Please review the figure legends, as the panel orders are incorrect in several figures, such as Figure 3, Figure 4, Figure 5, Figure 6, and Figure 7. Kindly correct them.
Response: Thanks for your advice.We have corrected the panel orders of Figure 3, Figure 4, Figure 5, Figure 6 and Figure 7.
8. In the case of Figure 7, a new type of statistical significance annotation appears, which differs from the previous figures and is not consistent with the figure legends. I recommend using the same scheme for Figure 7 as in the other figures throughout the MS.
Response: Thanks for your comments, and we have completed the revision according to your intention (page 19, line 509).
9. Please provide information on what type of kit was used for the measurement of MPO activity. Also, in line 292, the Authors state that "the expression of MPO activity". Please clarify if the activity or expression was measured.
Response: Thanks for your advice.We have provided information about the type of kit used for measurement of MPO activity (page 4, line 134-135). In addition, we have modified " the expression of MPO activity " to " the MPO activity "(page 11, line 322).
10. Please add visible scale bars to the H&E and TEM images. It would be beneficial if the specific alterations were marked on the H&E-stained images with arrows (distortion of crypt architecture, thickening of the muscularis mucosae, etc.). Please also include the magnification in the figure legend.
Response: Thanks for your advice.We have added a visible scale and indicated the magnification in the figure legend. In addition, specific alterations were marked on the H & E -stained image with arrows. ( red arrow represent inflammatory cell infiltration, green arrow represent crypt integrity destruction ) (page 9, line 270).
11. While analysing the NF-κB pathway, did the authors assess whether IκB expression or phosphorylation was affected?
Response: We appreciate your suggestion, we have supplemented the data of IκB mRNA levels . (Supplementary Figure 10, 11 and 12). The updated results show that SPS significantly inhibits IκB phosphorylation, further confirming its regulatory effect on the NF-κB pathway.
12. I suggest reconsidering the summary figure, as it is currently difficult to interpret and contains too much data. Instead of presenting individual results, it would be more effective to summarise the key conclusions drawn from the findings.
Response: We appreciate your suggestion, the optimized summary chart has been replaced at the end of the paper. We believe that the revised chart can convey the core conclusions of the research more clearly.
Reviewer 3 Report
Comments and Suggestions for Authors
The paper entitled: “Safflower polysaccharides alleviate ulcerative colitis by modulating gut immunity”, shows an interesting approach and demonstrates its therapeutic potential in the treatment of ulcerative colitis, although the manuscript is well written and complete, minimal comments are suggested to improve it.
Paragraph 87, 97, 181, 185, missing space between words, revise entire document.
Section 2.4, please include information on the acceptance of the project by the ethics committee in the use of institutional animals, for example, the folio, as well as the total number of animals used.
Include in this same section the concentration or dose administered to the animals, volume and route of administration used. In the image include in the figure caption the description of the abbreviations, correct the units of the SPS group. Also include the description of the model and the induction in the animals.
In Figure 1E, include in the images aided with arrows or some pointing of the relevant structures and changes in the tissues, as well as in the figure caption, as described in the text.
Comments on the Quality of English Language
No more comments
Author Response
1. Paragraph 87, 97, 181, 185, missing space between words, revise entire document.
Response: Thank you very much for pointing out the formatting issues ! We have carried out a comprehensive review of the manuscript, corrected space missing, to ensure that the language is clear and accurate, standardized and unified format. And here we did not list the changes but marked in red in the revised paper.
2. Section 2.4, please include information on the acceptance of the project by the ethics committee in the use of institutional animals, for example, the folio, as well as the total number of animals used.
Response: Thanks for your comments,we have refined the animal ethics statement and placed it in Section 2.4 (page 4, line 173).
3. Include in this same section the concentration or dose administered to the animals, volume and route of administration used. In the image include in the figure caption the description of the abbreviations, correct the units of the SPS group. Also include the description of the model and the induction in the animals.
Response: Thanks for your comments, we supplemented the complete description of the dose and route of administration in the original experimental design, supplemented the units of the SPS group, and added the detailed construction process of the model in the animal experiment in Section 2.4 (page 4, line 164-169). Finally, we explain the abbreviations in the figure caption.
4. In Figure 1E, include in the images aided with arrows or some pointing of the relevant structures and changes in the tissues, as well as in the figure caption, as described in the text.
Response: Thanks for your advice,according to your suggestion, we have marked specific alterations were marked on the H & E -stained image with arrows, ( red arrow represent inflammatory cell infiltration, green arrow represent crypt integrity destruction ) and explained them in the figure caption (page 9, line 270).
Reviewer 4 Report
Comments and Suggestions for Authors
Authors have submitted a research article titled Safflower polysaccharides alleviate ulcerative colitis by modulating gut immunity. This research article has significant interest and provides valuable information about the lipopolysaccharide anti-inflammatory activity in both in vitro and in vivo models. This article may be suitable in your journal after necessary revisions.
The manuscript does not mention an ethical approval number for the animal experiments. This information must be provided to ensure compliance with institutional and international ethical standards (All experiments were con-151 ducted in accordance with the principles outlined in the *Guide for the Care and Use of 152 Laboratory Animals* (National Institutes of Health, USA) and were approved by the Med-153 ical Ethics Committee of Shihezi University).
First, authors have stated that Our previous studies confirmed that SPS ameliorates ulcerative colitis by restoring the intestinal mucosal barrier and modulating gut microbiota (Line Nos. 464–465). Similar experiments (Safflower polysaccharide ameliorates acute ulcerative colitis by regulating STAT3/NF-κB signaling pathways and repairing intestinal barrier function) were conducted and reported previously. Authors must clarify the novelty of the current research data compared to previously published articles.
The authors have reported several scientific data at the clinical and molecular levels. However, microbial modulation in UC is not analyzed. It would provide stronger evidence if the authors could investigate the microbiome changes in experimental animals because the microbiome plays a key role in inflammatory responses.
An explanation is required for abbreviations when they are first used, particularly in the abstract.
The experiments using siRNA-CHI3L1 and r-CHI3L1 elucidated that CHI3L1 mediates the interaction between epithelial cells and immune cells. These lines are not clear.
Table 3 is unnecessary; either delete it or move it to the supplementary file. I recommend the authors provide only the details such as antibody name and company.
Table 4 should also be moved to the supplementary file.
Interpretation for Figure 2 AE is missing.
Cytotoxicity assays demonstrated that TNF-α concentrations ranging from 10 to 200 ng/mL maintained cell viability >80%, confirming the absence of significant cytotoxicity within this range (Figure 4A). Subsequent evaluation confirmed that SPS exhibited no significant cytotoxicity within the concentration range of 10–320 μg/mL (cell viability >80%) (Figure 4B). These statements are not clear; make them clear and understandable.
There is a spelling mistake in Figure 4B X-axis. Please check and revise it.
Authors must move Figures 6 and 7 to the respective places in the results section.
Figure 1 legend: Rats. (A) The body weight changes of rats. (B) Body weight on the 7th day. (C) DAI score. (D) Colon length. (E) The H&E staining of colon. (G) The TEM imaging of colon. There is a labeling mistake—Figure 1 (G) is referenced but not labeled in the actual figure content provided.
The selection of 10, 40, and 80 µg/mL SPS doses in vitro seems empirical. Were these based on pharmacological relevance or cytotoxicity only?
Authors are requested to provide clear figure legends for each figure, including details such as simple experimental procedure, explanation for all abbreviations, group details, dose treatment, and incubation periods, to make them easy to understand for readers. Just revise and give me in similar formats.
Author Response
1. The manuscript does not mention an ethical approval number for the animal experiments. This information must be provided to ensure compliance with institutional and international ethical standards (All experiments were con-151 ducted in accordance with the principles outlined in the *Guide for the Care and Use of 152 Laboratory Animals* (National Institutes of Health, USA) and were approved by the Med-153 ical Ethics Committee of Shihezi University).
Response: Thanks for your comments,we have refined the animal ethics statement and placed it in Section 2.4 (page 4, line 173).
2. First, authors have stated that Our previous studies confirmed that SPS ameliorates ulcerative colitis by restoring the intestinal mucosal barrier and modulating gut microbiota (Line Nos. 464–465). Similar experiments (Safflower polysaccharide ameliorates acute ulcerative colitis by regulating STAT3/NF-κB signaling pathways and repairing intestinal barrier function) were conducted and reported previously. Authors must clarify the novelty of the current research data compared to previously published articles.
The authors have reported several scientific data at the clinical and molecular levels. However, microbial modulation in UC is not analyzed. It would provide stronger evidence if the authors could investigate the microbiome changes in experimental animals because the microbiome plays a key role in inflammatory responses.
Response: Thank you very much for your comments. In our previously published work from the same research group, we employed a DSS-induced ulcerative colitis mouse model as the research object. In contrast, the present study utilizes a TNBS-induced colitis rat model. This difference in research subjects (mouse vs. rat) and experimental models (DSS-induced vs. TNBS-induced colitis) constitutes a key novelty of the current work: it verifies that SPS exerts a stable regulatory effect on the STAT3/NF-κB signaling across different animal species and distinct colitis induction models. Such verification further supports the reliability of SPS’s therapeutic effect on colitis and provides more comprehensive preclinical evidence for its potential clinical application.
Thank you for this valuable suggestion. We fully agree that investigating changes in the gut microbiome is crucial for validating the role of SPS in ulcerative colitis UC, given the microbiome's key involvement in inflammatory responses. Our research team has planned to incorporate microbiome-related experiments into the subsequent phase of this study, which will be conducted by other members of our group.
3. An explanation is required for abbreviations when they are first used, particularly in the abstract.
Response: Thanks for your comments, we have comprehensively sorted out the use of abbreviations in the full text and implemented the revisions one by one.
4. The experiments using siRNA-CHI3L1 and r-CHI3L1 elucidated that CHI3L1 mediates the interaction between epithelial cells and immune cells. These lines are not clear.
Response: Thanks for your comments, we have supplemented and improved the content of siRNA-CHI3L1 and r-CHI3L1 experimental demonstration of CHI3L1-mediated epithelial-immune cell interaction : In the original text , a description was added - “ That was to say. overexpression of CHI3L1 in epithelial cells increased the proportion of M1 polarization in immune cells and promoted the secretion of pro-inflammatory factors such as IL-6. TGF-β and TNF-α ' clear this experimental conclusion.” (page 18, line 475-477 ).
5. Table 3 is unnecessary; either delete it or move it to the supplementary file. I recommend the authors provide only the details such as antibody name and company.Table 4 should also be moved to the supplementary file.
Response: Thank you for your comments, we have adjusted according to the opinions : delete Table 3, only the antibody name and company are provided in the ' 2.1 experimental reagent ' section; table 4 has been moved to the supplementary document ( labeled ' Supplementary Table S1 ' )
6. Interpretation for Figure 2 AE is missing.
Response: Thank you for your correction , we have supplemented the interpretation in the original text to Figure 2.
7. Cytotoxicity assays demonstrated that TNF-α concentrations ranging from 10 to 200 ng/mL maintained cell viability >80%, confirming the absence of significant cytotoxicity within this range (Figure 4A). Subsequent evaluation confirmed that SPS exhibited no significant cytotoxicity within the concentration range of 10–320 μg/mL (cell viability >80%) (Figure 4B). These statements are not clear; make them clear and understandable.
Response: Thank you for your comments, we have optimized the relevant content and revised it to :The results of cytotoxicity experiments showed that the cell survival rate remained above 80 % in the concentration range of 10-200 ng / mL of TNF-α, indicating that the concentration range did not cause significant cytotoxicity ( Figure 4A ). Similarly, when the concentration of SPS was 10-320 μg / mL, the cell survival rate also remained above 80 %, further confirming that the drug did not have significant cytotoxicity in this concentration range ( Figure.4B ) (page 12, line 349-355).
8. There is a spelling mistake in Figure 4B X-axis. Please check and revise it.
Response: Thank you for your correction , we have revise the spelling of the X axis of Figure 4B.
9. Authors must move Figures 6 and 7 to the respective places in the results section.
Response: Thank you for your optimization suggestions ! We have moved Figure 6 and Figure 7 to the corresponding discussion in the results section.
10. Figure 1 legend: Rats. (A) The body weight changes of rats. (B) Body weight on the 7th day. (C) DAI score. (D) Colon length. (E) The H&E staining of colon. (G) The TEM imaging of colon. There is a labeling mistake—Figure 1 (G) is referenced but not labeled in the actual figure content provided.
Response: Thank you very much for pointing out the problem of Figure 1 ! We have confirmed that the ( G ) item mentioned in the legend of Figure 1 has been changed to the ( F ) item to ensure that the legend is completely corresponding to the content of the graph.
11. The selection of 10, 40, and 80 µg/mL SPS doses in vitro seems empirical. Were these based on pharmacological relevance or cytotoxicity only?
Response: Thank you for your comments, in this study, the selection of 10,40,80 μg / mL SPS doses in vitro experiments was not only based on cytotoxicity, but also on pharmacology-related experiments. The specific basis for the dose selection is dependent on the QRT-PCR experiment. The specific experiment is shown in Supplementary Figure 7.
12. Authors are requested to provide clear figure legends for each figure, including details such as simple experimental procedure, explanation for all abbreviations, group details, dose treatment, and incubation periods, to make them easy to understand for readers. Just revise and give me in similar formats.
Response: Thank you for your comments, we have added clear legends to each picture, and added brief information about the experimental procedure, grouping details, dosage(page 4, line 164-169) and incubation time(page 14, line 398) to the original text, and explained abbreviations.
Round 2
Reviewer 2 Report
Comments and Suggestions for Authors
The authors were very cooperative and addressed all of my questions. The manuscript has improved.
Author Response
Dear reviewer :
We sincerely thank you for completing the second round of review of this manuscript ' Safflower polysaccharides ulcerative colitis by modulating gut immunity ' in your busy schedule, and give you the affirmation of agreeing to publish.
Your previous professional suggestions have provided key help for us to improve manuscripts and improve the quality of research presentation. This recognition is not only an encouragement to our research work, but also allows us to see more clearly the academic value of the results.
In the future, we will actively cooperate with the journal to complete the publication process, and once again extend our sincere thanks to you !
Wish you a smooth work !
Reviewer 4 Report
Comments and Suggestions for Authors
I have carefully reviewed the revised manuscript entitled “Safflower polysaccharides alleviate ulcerative colitis by modulating gut immunity.” The authors have thoroughly addressed all of my previous comments and have made the necessary revisions. I find the current version of the manuscript to be scientifically sound and suitable for acceptance in your journal.
However, I note that the plagiarism report submitted by the authors indicates a similarity rate of 32% report submitted by the authors. it would be appreciated if the authors could make further efforts to reduce the similarity index to below 20%, or to a level acceptable by the journal. This will help ensure the originality and integrity of the manuscript.
Author Response

(The authors gave the same response as above.)
